# Recent Advances in the Fabrication and Application of Graphene Microfluidic Sensors

**DOI:** 10.3390/mi11121059

**Published:** 2020-11-30

**Authors:** Shigang Wu, Xin Wang, Zongwen Li, Shijie Zhang, Fei Xing

**Affiliations:** 1School of Materials Science and Engineering, Shandong University of Technology, Zibo 255049, China; wshgang@sdut.edu.cn; 2School of Physics and Optoelectronic Engineering, Shandong University of Technology, Zibo 255049, China; wx15550343721@163.com (X.W.); zhangsj.sdut@foxmail.com (S.Z.)

**Keywords:** graphene, microfluidics, electrochemical, optical, biosensors

## Abstract

This review reports the progress of the recent development of graphene-based microfluidic sensors. The introduction of microfluidics technology provides an important possibility for the advance of graphene biosensor devices for a broad series of applications including clinical diagnosis, biological detection, health, and environment monitoring. Compared with traditional (optical, electrochemical, and biological) sensing systems, the combination of graphene and microfluidics produces many advantages, such as achieving miniaturization, decreasing the response time and consumption of chemicals, improving the reproducibility and sensitivity of devices. This article reviews the latest research progress of graphene microfluidic sensors in the fields of electrochemistry, optics, and biology. Here, the latest development trends of graphene-based microfluidic sensors as a new generation of detection tools in material preparation, device assembly, and chip materials are summarized. Special emphasis is placed on the working principles and applications of graphene-based microfluidic biosensors, especially in the detection of nucleic acid molecules, protein molecules, and bacterial cells. This article also discusses the challenges and prospects of graphene microfluidic biosensors.

## 1. Introduction

The rapid development of Microfluidics Technology in recent years has caused a revolutionary impact in the fields of physics [1,2], materials [3,4], and biomedicine [5,6,7]. The technology includes the control, operation, and detection of complex fluids at microscopic dimensions, with particular emphasis on the construction of microfluidic channel systems to achieve various complex microfluidic manipulation functions [8,9]. In comparison to macroscopic systems, microfluidic devices offer advantages in high-throughput, homogeneous reactions, shorter time to results, high sensitivity [10], and also demonstrate the commercial application value of low cost [11], low energy consumption, repeatability, and highly integrated multi-function [12], etc. In recent years, there have been considerable efforts have been made to combine the sensors with microfluidics to further broaden the applications of sensors [13]. As the crucial device in the microfluidic control system, different from traditional sensors, microfluidic sensors have the advantages of high sensitivity, small size, real-time monitoring, accurate measurement, and easy matching with microfluidic equipment [14,15,16] Microfluidic sensors are not only small in size, but also can be used as an analysis system platform (see Figure 1) [17], combined with the outstanding characteristics of nanomaterials to make it have better selectivity and sensitivity, and has attracted widespread attention in a host of fields [18,19,20].

Due to the reactivity, surface and interface effects, quantum size effect, Raman spectrum effect, catalytic efficiency, electrical conductivity, and other characteristics of nanomaterials, nanomaterials have excellent applications in the field of microfluidics technology [21,22]. Graphene, as the optimum material, to construct the microfluidic devices among numerous nanomaterials due to its unique structural characteristics and excellent performances [23,24], such as the nanostructure of the honeycomb arrangement of carbon atoms, as well as excellent physical and chemical properties [25,26]. The signal layer graphene has better transparency it can penetrate 98 percent of visible light, it has ultra-high electron mobility and thermal conductivity, ultra-thin (about 0.35 nm) ultra-light structure, low planar density (0.77 mg/m^2^) [27,28]. The special two-dimensional (2D) structure of graphene, in which each carbon atom is exposed to its surface, makes it sensitive to changes in the charge environment caused by the adsorption of a target [29,30,31]. The surface of graphene and its oxide materials has active functional groups due to the intercalation of carboxyl, hydroxyl, and epoxy ions, thereby greatly improving the cell and biological reactivity of the material [32,33,34]. The biocompatibility of graphene has further broadened its application to biomedical fields such as the clinical diagnosis and drug delivery using optical and electrochemical sensors [35,36,37]. Based on these special properties, graphene combined with basic microfluidic technology has been widely used in microfluidic chips and devices [28,38]. Besides, graphene derivatives, such as graphene oxide (GO), reduced graphene oxide (rGO) and functional graphene, are also ideal materials for constructing microfluidic channels [39,40].

The combination of graphene and microfluidic sensors can amplify their respective strengths to enable even more valuable and potential applications. Santangelo et al. [41] developed a microfluidic sensing platform for detecting low-concentration analytes, even low-concentration toxic heavy metals, based on the advantage that graphene is extremely sensitive to foreign matter. The sensor is not only simple and accurate to develop, but also has a strong sensitivity to the lowest concentration of analytes. Bouilly et al. [42] designed a nanoelectronic biomolecular sensor based on the combination of a graphene field-effect transistor (GFET) device array and a microfluidic circuit for detecting DNA oligonucleotide sequences or antibody-antigen coupling [43]. The devices can exhibit the electrical characteristics of specific and non-specific biomolecule interactions with graphene materials, and lay the foundation for electronic detection of breast cancer and leukemia biomarkers for clinical diagnostic applications [44,45]. Numerous studies have shown that it is possible to develop miniature microfluidic sensors with remarkable performance owing to the size and unique properties of graphene materials, which opens up new opportunities in the field of sensor analysis and detection [46,47].

The purpose of this article is to provide a comprehensive overview from production to the application of graphene microfluidics devices and the latest progress of graphene microfluidics sensors in recent years. In Section 1 we briefly introduce the structure and properties of graphene as well as the features and advantages of microfluidics devices. The unique advantages of the graphene microfluidics biosensors are also discussed. Fabrication of the graphene microfluidics devices is presented in Section 2. Here we first introduce the preparation of graphene and the functionalization of graphene. In Section 3, the current fabrication process and materials of microfluidic chips will be highlighted. It focuses on the latest application of graphene microfluidics devices in biology, optics, and electricity in Section 4. Finally, the conclusions of this article are summarized, and the challenges and application prospects of graphene microfluidic sensors are also proposed.

## 2. Preparation and Functionalization of Graphene and Its Derivatives

### 2.1. Preparation of Graphene and Its Derivatives

Fabrication methods of graphene may be divided into top-down and bottom-up two main categories.

#### 2.1.1. Top-Down Approach

The top-down method is mainly exfoliation (like micromechanical, electrochemical, thermal), and also includes the reduction of GO, sonication, etc. Micromechanical exfoliation is the first method used to obtain graphene with different layers from graphite [48]. Recently, Sinclair et al. [49] carefully considered the intermolecular interaction of graphene, and designed a process for stripping graphene from polymer tape, as shown in Figure 2a. Graphite containing multiple graphene sheets is compressed between two polymer layers under isothermal-isostatic conditions. Exfoliate the graphite by increasing the height between the polymer layers at a constant speed, thereby high-quality graphene is obtained in the canonical ensemble [50]. Although this method is usually widely used to prepare high-quality graphene samples for analysis, it is not suitable for commercial production due to its time-consuming and labor-intensive shortcomings [51]. After that, with the development of technology, there gradually emerge electrochemical and thermal exfoliation techniques. Electrochemical peeling is similar to liquid phase peeling but applying electric field force to drive electrolyte molecules directly into the graphite cathode electrochemically [52]. In this way, the van der Waals (vdW) forces between the graphite layers become weaker with the increase of the layer spacing. Therefore, it is easy to obtain graphene by directly electrochemically exfoliating graphite sheets [53]. Moreover, compared to mechanical exfoliation, the electrochemical exfoliation approach makes mass production of high-quality graphene easier. Graphene prepared by electrochemical exfoliation possesses better physical and chemical properties because the layer structure is not damaged. During the preparation process, due to the strong electrochemical electric field, the peeling efficiency is higher, and by accurately controlling the current and voltage, it is easy to realize the controllable preparation and performance regulation of graphene. Additionally, since GO is easier to exfoliate than graphite, graphene is usually made from GO by chemical reduction. The preparation of GO is usually carried out using graphite, oxidising agents and concentrated acid raw materials via Hummers, or the redox method [54]. Then, the GO is exfoliated by heat treatments or the acoustic in water, and finally, the graphene can be obtained via the reduction of GO using the chemical or thermal methods. Due to the final product is not completely reduced in the reaction process, it is known as the rGO instead of graphene. However, the explosive and toxic chemical reducing agents limit the large-scale production of graphene [55]. In recent years, various biomolecules have been widely used to synthesise graphene and its derivatives because of the availability of them. Microbial reduced graphene approach, GO reduction by bacterial respiration, shows great electrochemical quality [56]. And it also has obvious advantages in terms of inexpensive, timesaving, good biocompatibility and non-toxic compared with additional electrochemical and mechanical methods [57].

#### 2.1.2. Bottom-Up Approach

The bottom-up approach is to synthesize graphene using small carbon-containing molecules as raw materials, which mainly includes chemical vapor deposition (CVD), pyrolysis of SiC, and crystal epitaxy. The high purity and crystalline graphene can be obtained by all of these techniques. Nowadays, the CVD method has proven to be the most widely used fabrication technique for obtaining high quality and large area graphene due to process tunability and mass production. The growth of graphene by CVD can in principle be divided into two types, that is, the carburizing mechanism and the surface growth mechanism. The difference between the two mechanisms lies in the carbon content of the metal substrate. CVD methods use carbon compounds such as methane as carbon sources to grow graphene by decomposition at high temperature on the substrate. The growth of graphene on Cu metal surfaces by CVD processes, which reduce the decomposition temperature and graphitisation temperature of carbon precursors, is now being extensively studied. The growth of graphene on Cu, Ni and Cu/Ni surfaces was investigated by Li’s group [58] using carbon isotope labelling techniques, as shown in Figure 2b,c. In the CVD process, due to the catalytic activity of the metal surface and the high solubility of carbon, the decomposition of methane provides ^12^CH_4_ or ^13^CH_4_, which rapidly diffuses into most metals. Then, when the carbon in the metal reaches supersaturation at a certain temperature, due to carbon segregation, balanced graphene will form on the surface. Moreover, the carbon concentration and the cooling rate directly determine the number of layers of graphene on the metal surface. Besides the metal substrates, the synthesis of graphene on the insulating substrates has also been studied to be feasible. Tai et al. [59] conducted vdW epitaxial growth of graphene on a single crystal silicon substrate placed upside down by metal-free atmospheric CVD at 900–930 °C, as shown in Figure 2d. Due to the catalytic inertness of silicon, the thermal decomposition of methane produces activated carbon that triggers the nucleation of graphene. The high saturation and collision frequency of carbon radicals can enhance the nucleation of graphene when the substrate is upside down. The methane under the flux of 180 sccm continuously decomposes into activated carbon, leading to the subsequent growth of graphene. At higher temperatures, as the silicon surface becomes active, graphene begins to spread and grow around its edges, resulting in the formation of concave double-layered regions of larger size. The excess activated carbon also begins to nucleate in the core of the domain. At this point, a bulging oligomeric graphene domain is generated during core propagation. Improved growth conditions, like the CH_4_ partial pressure or H_2_/CH_4_ ratio, were also experimentally investigated. In short, it is possible to deposit multiple layers of graphene on an insulating substrate without catalysis. Furthermore, to manufacture high-quality graphene, a higher decomposition temperature is required on an insulating substrate than on a metal substrate including Cu or Ni.

### 2.2. Functionalization of Graphene and Its Derivatives

Although graphene exhibits great application potential in many fields due to its excellent physical and chemical properties, it is worth mentioning that the strong vdW force between graphene sheets is far greater than the interaction with solvents, making it difficult to dissolve in water and common organic solvents, which also limits its wide application in microfluidic platforms. This is also the biggest obstacle to the application of graphene as a sensing medium layer to sensor devices. Hence, it has greatly significant to modify the characteristics of graphene and expand the application of it in the function of nanoelectronic devices especially in the microfluidics field via functionalization. Although graphene has a stable hexagonal structure and chemical inertness, the actual graphene produced has some defects and its edges are very active, which provides the possibility to realize the functionalization of graphene. One of the most effective methods to functionalize graphene is to reduce or peel the graphite with an alkaline metal in an appropriate solvent and then the intermediate graphene formation is quenched with an electrophilic body. Besides, the functionalization of graphene can avoid agglomeration by producing the strong polar-polar interaction of hydrophilic. The functionalisation of graphene covers many areas of research, including the chemical modification of its surface, reactions with various molecules, and covalent and non-covalent interactions [60,61,62]. The method currently adopted is mainly to carry out effective and controllable functional modification of the graphene surface. There are two main methods for functional modification of graphene, i.e., non-covalent bond and covalent bond functional modification. Table 1 shows many differences between covalent functionalization and non-covalent functionalization, including theory, type, application, etc.

#### 2.2.1. Covalent Functionalization of Graphene

Covalent bond functionalization of graphene is the most widely studied functionalization method. After the edges and defects of graphene are oxidized, the surface contains a large number of active epoxy groups, such as hydroxyl groups, carboxyl groups, etc., so it can be covalently modified by a variety of chemical reactions. It not only can increase the solubility of graphene but also can offer new properties by introducing the organic functional groups. There are two main routes to achieve the organic covalent functionalization of graphene. One is by forming covalent bonds between free radicals or dienophiles and C=C bonds of graphene, the other is by forming covalent bonds between organic functional groups and the oxygen groups of GO. Free radicals, atoms or groups of unpaired electrons, are highly reactive and react with sp^2^ carbon atoms of graphene to form covalent bonds. The common free radicals are aryl diazonium salts and benzoyl peroxide, which are synthesized by Bergman cyclization and Kolbe electrosynthesis. In addition to free radicals, dienophiles also can react with the sp^2^ of graphene. For example, one of the most common dienophiles Azomethine ylide, it has been successfully used to functionalize carbon nanostructures (fullerenes, nanotubes, and nanohorns) via 1, 3 dipole ring addition reaction. The dihydroxyl phenyl group was decorated on a graphene sheet with pyrrolidine rings, which was formed perpendicular to the graphene surface by the addition of azomethine ylide precursors (see Figure 3a) [65]. The azomethine ylide can be obtained by the condensation reaction between the 3,4-dihydroxybenzaldeyde and sarcosine. Introducing the hydroxyl groups into graphene increase the dispersibility of graphene in the polar solutions including ethanol and *N*,*N*-dimethylformamide. Moreover, by comparing the I_D_/I_G_ ratio and peak curve before and after the function of graphene in Figure 3b, it can be a conclusion that the functionalization of graphene causes a significant increase in sp^3^ planar carbon atoms. There are also many methods that can achieve the functionalization of graphene such as atomic radical addition, nucleophilic addition, cycloaddition, and electrophilic substitution reaction.

The functionalised graphene sheets are further modified by chemical reactions such as surface polymerisation, ion reduction and amidation to achieve higher chemical and thermal stability. At the meantime, functional graphene sheets exhibit electrical conductivity and represent outstanding processability and dispersibility in solutions. By contrast to graphene, GO contains a large number of radical energy groups, causing it to achieve covalent functionalization through various chemical reactions. For example, it usually is used as the starting material for the fabrication of graphene derivatives by the covalent attachment of organic groups on its surface. There are still several oxygen groups and defects after any reduction treatment of GO in the experiments. Hence, it can achieve that the added groups are linked via the oxygen atoms of GO in this type of functionalization. And there also remains a large number of oxygen groups after the functionalization because no further structural perturbations have occurred [67]. The covalent bond function of graphene greatly improves its processing properties, giving graphene and its oxides some new properties. However, the covalent bond functionalization of graphene also has inevitable shortcomings. Covalent bond modification of graphene will destroy the intrinsic structure of graphene and change its own unique physical and chemical properties. The introduction of molecules and polymers with specific functions at the edges and defects of graphene will be the main research trend for the functionalization of graphene covalent bonds in the future.

#### 2.2.2. The Non-Covalent Functionalization of Graphene

In addition to functionalization with covalent bonds, non-covalent bonds (such as π-π interactions, ionic bonds, and hydrogen bonds) can also be used to modify molecules to functionalize the graphene surface and form a stable dispersion system. It is essential to increase the solubility and avoid agglomeration of graphene by the non-covalent functionalization with different organic compounds. Moreover, as a hydrophobic material, graphene must be non-covalently functionalized to make it soluble in polar solvents. To achieve the non-covalent bonding, there may have multiple routes including π-interaction, electrostatic, hydrophobic, vdW interactions. Among them, it must be said that π-interaction is an attractive method for graphene functionalization because it can connect functional groups to graphene without disturbing the electronic network of graphene.

As a π-system, non-covalent molecular interactions involving graphene are essential for stabilizing proteins, nucleic acid molecules, inorganic molecules and functional nanomaterials [68]. Due to the tiny change that happened in the electronic properties of π-systems can cause tremendous effect in the structure and characters of the nanosystem, the most relevant applications of interactions contain π-systems are about the fabrication of nanodevices and nanomaterial design. The π-interactions can be further divided into H-π interaction, π-π interaction, cation-π interaction, and antion-π interaction [69]. In process of π-interactions, there are both attractive forces like static electricity and dispersion, as well as repulsive forces. Each of these components differs from magnitude, physical origin, and directionality but the strength of the π-interactions are determined by the combined effect of them.

The H-π interaction, one of the hydrogen bonds, plays an important role in adjusting the geometry and the nature of the complex. Additionally, the substantial contribution provides by dispersion can further stabilize the H-π complexes. At present, the π-π interaction is the most attractive non-covalent interactions, that is, the negatively charged and diffused electron cloud of the π-system exhibits attractive interactions. For example, the dispersion interactions also take hold when two π-system possess the same electron densities in noncovalent π-π interactions [70]. However, when the systems hold different electron densities including one of the systems are electron-rich and the other is electron-deficient, the resulting complexes of the system bind through induction interactions [71]. The attraction can be reflected by the interaction energy in the experiment, and it has a significant impact on the interaction of the phenyl rings in the solution environments. In surface-substituted aromatic systems, the electron density of the parasite is a crucial stabilizing factor, which not only affects the electron donating ability responsible for static electricity but also affects the dispersion interaction and exchange repulsion. Besides, dispersion energy and exchange repulsion will in turn increase the electrostatic energy of surface-substituted aromatic systems. Kim et al. [72] has developed and synthesised a new molecular system that exploits differences in non-bonding interactions to exhibit a motion during redox processes. The photochemically and electrochemically active π-system was utilized to have maximum control of this interconversion. The π-π interaction is one of the important ways to realize supramolecular self-assembly. Therefore, the design and synthesis of novel organic nanostructures can be achieved by controlling the relationships of several non-covalent interactions. Graphene is very flexible and less expensive, but the energy conversion rate of it is not high. Wang et al. [66] proposed a noncovalently modulated graphene film using the pyrene butanoic acid succidymidyl ester (PBASE) to further improve the energy conversion rate. The inset of Figure 3c shows that the PBASE molecules are attached to the surface of a graphene sheet by means of π-π interactions. The effect of the π-π interaction between the graphene and PBASE on the optical absorption of visible graphene films can be negligible, as shown in Figure 3c. And compare with the pristine graphene, the power conversion efficiency of functionalized graphene was improved to 1.71%. As can be seen from Figure 3d, the UV and reflected photoelectron spectra are very similar at the Fermi energy, and graphene maintains a near zero-gap state both before and after modification with PBASE. In the π-interactions, the cation-π interaction is enhanced by the electrostatic and attraction energy between the metal cation and π [73], which is also superior to π_cation_-π interactions, so that various receptors with strong binding energies and high selectivity for metal cations can be explored. The π_cation_-π interactions also exhibit different advantages to metal cation-M interactions, such as a stronger total binding energy.

## 3. The Fabrication Processes and the Materials of Microfluidics Devices

For the microfluidics field, one particularly attractive idea is to develop integrated “lab on a Chip (LOC)” systems that can reproduce laboratory-scale processes in a simplified way that costs less, takes less time, and takes up less space than traditional equivalents. Besides, graphene and its derivatives (like GO, rGO, and functionalized graphene, etc.) are very suitable for the basic technical functions of microfluidics, so they have been widely used in LOC devices.

### 3.1. The Fabrication Processes of Lab-On-A-Chip Devices

As a typical representative technology of a LOC, microfluidic technology has developed rapidly. The microfluidic chip is the main platform for the realization of microfluidic technology. The main feature of the device is that the effective structures (channels, reaction chambers, and some other functional components) containing fluids are at least micron-sized. Besides, in the microfluidic-based LOC devices, fluid transport is carried through laminar co-flow or immiscible flow in a segmented flow within a small channel. Therefore, the fabricating process of the microfluidic chip is particularly important. It has been reported that many microfluidic fabricating processes have been proposed for various materials and applications. Under the major fabricating processes, many sub-processes including etching, lithography, thermoforming, hot embossing, polymer casting, and bonding are used individually or collectively to implement the final form of LOC devices. Among them, hot embossing and bonding are the most widely used methods in microfluidic chip processing. Glass microfluidic chip prepared by hot embossing have great potential in the fields of medical detection, drug analysis, air monitoring and optical sensors. Jiang et al. [74] fabricated glass microfluidic channels through a novel thermal embossing strategy that can also be used for microfabrication of other amorphous materials, as shown in Figure 4a. In this study, a glass mold inserts with a high transition temperatures fabricated in the supercooled liquid region play an important role for glass embossing with low transition temperatures. High transition temperature glass mold structures with excellent high temperature resistance and thermoformability can be produced by controlling processing conditions such as embossing temperature, pressure and duration. The shape transferability of the low transition temperature glass microfluidic channel reaches ~95%. The proposed method has feasibility and versatility in the preparation of microfluidic chips. However, the hot pressing method also has the disadvantage of high heating temperatures and pressures, which affect the precision of the microstructure and make it unable to fully meet all the requirements for preparing of microfluidic chips. Bonding is to pattern all the features on the substrate and then glue it with the cover to create a closed microchannel for fluid analysis. Microfluidic chip bonding methods mainly include hot pressing and adhesive. Kurihara et al. [75] proposed a new low-deformation thermal bonding method for manufacturing a single polymer material chip, which uses two different compounds of the same polymer material with different transition temperatures and nanostructured plates to achieve bonding with low deformation accuracy. For low-temperature bonding, the deformation rate of the polystyrene board chip with nanostructure is only 1.1%, which is significantly smaller than the deformation produced by the ordinary thermal bonding process, so this bonding process can be used to replace direct thermal bonding and laser bonding, etc. The bonding failure rate by the thermal bonding method is high due to many factors, especially the lack of cleanliness and smoothness of the glass surface. To simplify the operation and improve the adhesion, Su et al. [76] proposed a fast bonding method to manufacture glass-based chips using a polyurethane (PU) composed of 8015-A (A-glue) and 8015-B (B-glue). The adjustable performance of the PU is affected by the distribution ratio of A-glue and B-glue. And after repeated tests, when the weight ratio of A-glue/B-glue is 3.6:1, it can be used to encapsulate glass chips. The results of many experiments have also proved the feasibility of the glass PU chip. This method of encapsulating chips using PU material has the lowest cost, the fewest steps, and the highest bonding success rate. The adhesive bonding method also has some drawbacks, such as generating a large number of bubbles, which can easily contaminate and block the microchannels. In this regard, Zhang et al. [77] investigated a new bonding method under pressure by UV-curing to fabricate microfluidic chips. Through the light-curing bonding method, the prepared microfluidic chip has high bonding strength and high speed, and the microchannel will not be contaminated by the adhesive. Therefore, permanent bonding can be achieved to effectively solve the problem of poor adhesion of microfluidic chips. The irreversibility of the chip is also an inevitable problem, which remains to be solved.

### 3.2. The Materials of Lab-On-A-Chip Devices

There are a wide variety of materials used to make microfluidics device, mainly includes silicon, metals, semiconductors and other inorganic materials, as well as polymers, hydrogels, paper and other organic materials [79]. The different kinds of materials hold significant advantages in some aspects and have played an important role in the corresponding field. In microfluidic chips, silicon materials are widely used and used as the main material of early microfluidic chips. Fornell et al. [80] fabricated 380 × 150 µm^2^ silicon channels with vertical walls and flat bottom surface in cross section by optimizing the power and pulse frequency of a nanosecond laser, which is a method for fabricating microfluidic channels in silicon using a laser system. The design of the microfluidic chip includes a resonance channel branched into a three-pronged outlet, and a glass wafer is used to seal the microfluidic channel using an adhesive. Due to its good electroosmosis, light transmission, surface biocompatibility, and processing technology similar to silicon materials, glass [81] is widely used as a substitute for silicon materials in microfluidic chips. Wang et al. [78] developed a Si-glass chip using a cyclic direct bonding method based on an oxygen plasma and annealing treatment process (see Figure 4b). The bonding process mentioned in this work has the advantage of high bond strength and a tight bonding interface to meet the needs of microfluidic devices. Besides, the bonding strength of the chip interface was verified by testing its corrosion resistance in various chemical and biological solutions, and the work of separating the bonding interface of the glass substrate in ethanol without cracking was explored.

Although glass has many advantages compared with other materials, the manufacturing cost of glass-based microfluidic chips is high and time-consuming and laborious, which limits its use. Polydimethylsiloxane (PDMS) is currently the most widely used polymer material in the field of microfluidic chips, and it is often combined with glass to make microfluidic chips. PDMS can be applied to biomedicine in the field of microfluidics, including on-chip devices and rapid real-time monitoring. Due to the enhanced signal response and ease of manufacturing, many microfluidic preparation processes use PDMS as basic material. Surface treatment, especially functionalization, has changed the physical and chemical properties, so it is aimed at a very wide range of sensing applications for PDMS-based microfluidic systems. The most important surface modification techniques commonly used in microfluidics are plasma and UV. Liu et al. [82] reported the preparation of a microfluidic channel in a highly innovative way, by sealing the control layer, liquid layer and the thin membrane of PDMS material together. Among them, the control layer and the liquid layer are manufactured by positive and negative photoresist molds, respectively. In this work, enhanced seal strength was achieved due to two factors, one being the use of an oxygen plasma surface treatment of the PDMS surface which helped to improve the properties and the other being the use of different ratios of PDMS prepolymer which helped to lower the bonding temperature and time. Besides, the entire sealing process was achieved at room temperature, which is convenient, simple and easy to perform. This method is more convenient than the one used for other commonly silicon-based materials (e.g., glass, quartz, etc.), and it also optimizes the PDMS-PDMS bonding process. To demonstrate its performance, the PDMS microfluidic chip maintains integrity under an applied pressure of 280 kPa provided by the N_2_ flow, which is perfectly suited to the practical needs of microfluidic chips. Olmos et al. [83] introduced a PDMS microfluidic device manufacturing method by manufacturing a photopolymer mold with a multilayer microstructure. In the experiment, the female photopolymer mold manufacturing method with multi-stage channels was successfully demonstrated by changing the cavity width and performing a reverse UVA exposure time.This method can obtain multiple molds with multiple microstructures in a unique part, where the mold has a minimum structural size of 10 mm and a structural height range of 53 to 1500 mm. Also, the thickness of the structure can be controlled by changing the channel width, thereby customizing the thickness according to the type of assay. The method for manufacturing PDMS microfluidic devices has many advantages, such as reduced manufacturing time, multiple structures with multiple topologies, multiple depths and heights in a single mold, and lower manufacturing costs. In the end, the PDMS microfluidic device developed to produce a hierarchical structure has great potential in different microbial fields, especially for cell culture and proliferation.

## 4. The Application of Graphene Microfluidics Devices

### 4.1. Graphene Field-Effect Transistor Sensors

Graphene, as a two-dimension material, has a large surface and high surface-area-to-volume ratio. On account of the special 2D structure, each carbon atom is exposed on graphene’s surface. Graphene is sensitive to changes in the environment caused by the adsorption of the analyte to its surface, in particular to the binding of organic or inorganic molecules on the surface [29]. Hence, it served as a promising material for the highly sensitive electrochemical sensors and also played an important role in the field-effect transistor (FET) field [84]. And there has a high electrode electron transfer rate on the edges and defects of it [85], which also turns its huge potential in electrochemical sensors [86]. For electrochemical sensors, the carbon nanotube is the material that is most widely utilized in electrocatalysis and the electrode [87]. But contrast to the carbon nanotube, the graphene-based electrode material has more advantages in the electrocatalysis activity. Most graphene electrochemical sensors used rGO because it can increase the electrochemical activity and enhance the performance of the sensor [88,89,90]. Besides, the advantages of rGO such as moderate and profitable preparation, high conductance, high surface-area-to-volume ratio [91], and tunable properties make it widely used in electrical sensors [88]. GO can immobilize the biomolecules by the covalent interaction as a tool. Hence graphene plays an important role in the electrochemical analysis [92].

Currently, based on the development trend of electrode surface functionalization and device module miniaturization based on electrochemical sensors, a new generation of nano-modified and integrated microfluidic technology on-chip detection system has been developed. For example, graphene is widely used as a working electrode in electrochemical detection platforms for the detection of heavy metals. The electrochemical analysis method is one of the most commonly used methods, because of the advantages of low power consumption, high sensitivity, short analysis time, and easy direct measurement [93]. In 2019, Santangelo et al. [41] reported a sensor (see Figure 5a) composed of 3D printed microfluidic chips and epitaxial graphene on SiC, which can detect heavy metals with high sensitivity and real-time. The sensor, with physical dimensions of 7 mm × 7 mm, is made by depositing four electrodes on the edges of the graphene surface. The four wires connecting the electrodes are soldered to the bottom of the SiC substrate, and then the sensor is constructed by applying an external force to fix the 3D printing chamber onto the top of the chip, as shown in Figure 5a. Two single-syringe pumps are used to inject the buffer solution and the analyte (Pb) into the mixed microfluidic chip, and then to the fluid cell for detection (see Figure 5b). Finally, the measurement is performed between the two angular contact points by biasing the sensor. In this study, the potential of the system for continuous monitoring of heavy metals was demonstrated by automatically injecting the reactants into microfluidic sensors to detect different concentrations of lead ions, as shown in Figure 5c. As the concentration of lead ions increases, the amount of charge transfer and hole conductivity between graphene and Pb^2+^ increases, but the Fermi energy level decreases. In the electrochemical microfluidic sensor, the sensing platform achieves significant sensitivity and low detection limits, although the area of graphene in contact with the solution is small. This work fully proves that the sensing platform can continuously monitor toxic heavy metals in real-time, and is convenient for direct sample research, which can be applied to daily life and industrial manufacturing. At present, it also shows some shortcomings, such as difficulty in determining the concentration of a single heavy metal in the presence of other heavy metals [94], defects in the preparation of uniform single-layer graphene, etc.

FETs are also one of the most widely used electrochemical microfluidic devices. Graphene has a suitable condition it was generally used as a channel material for FETs with applications in sensors and biosensors. Compared with other FETs devices made of classic semiconductor materials (such as silicon, GaN, SiC), GFETs show the development of flexible, stable, and biocompatible merits [95]. Many methods are fabricating GFETs sensors. Here, Wang et al. [96] developed a FET sensor with graphene channels for the direct measurement of hydroxyl radicals (•OH). The fabrication process of this graphene-based FET sensor is shown in Figure 5d, where a single layer of CVD-grown graphene is used as the sensing layer in the channel, while SiO_2_ and Si are used as the gate dielectric and back gate, respectively. The Cr/Au, which is patterned by photolithography, is located on both sides of the graphene surface and acts as an electrode. The design of the graphene channel FET sensor with the graphene/Au/Cys-PP structure is then completed by a sequence of operations involving the evaporation of gold nanoparticles (NPs), immersion in a cysteamine (Cys) solution and immobilisation of the protoporphyrin IX (PP). Figure 5e shows the schematic diagram of •OH detection. An unconfined chamber, approximately 8 mm wide, is assembled on the chip for the detection of all target solutions. After the Cd^2+^ aqueous solution is dropped into the chamber, the Cd^2+^ ions are combined with the doped graphene on the channel surface. After that, in the process of dropping the mixed solution that produces •OH, the I_ds_ monitored by the graphene/Au/Cys-PP-Cd^2+^ sensing layer shows a rapid response when corresponding to 1 × 10^−4^ M •OH within 2 s, as shown in Figure 5f. Quantitative metal ion doping can detect the •OH produced in aqueous solutions or living cells. Hence the FET sensor based on the graphene/Au/Cys-PP-Cd^2+^ structure enables real-time label-free detection of •OH and its concentration. Because of its label-free, high sensitivity, selective detection function, and miniaturization function, it has important value in human health and environmental monitoring.

In recent years, graphene-based FETs have attracted a lot of attention in various electrochemical and biosensor applications. Hence, a GFET is promising in the detection of various molecules, such as exosomes, bisphenol A (BPA), etc. For example, Yu et al. [97] have developed an rGO FET sensor for label-free electrical detection of exosomes with high sensitivity and specificity (see Figure 6a). In this work, the rGO FET sensor is fabricated by dropping the rGO solution prepared by the chemical reduction method on the induction channel of the chip, and then heating and annealing in a vacuum furnace. Figure 6b shows the working principle and operating procedure of the rGO FET-based biosensor for detecting exosomes. Among them, 1-Pyrenebutanoic acid succinimidyl ester plays a key role because its two ends are connected to the rGO surface and the CD63 antibody. When the exosomes flow through the sensor channel and bind to the CD63 antibody, the net carrier density on the chip surface changes due to the negative charge of the exosomes, which causes the Dirac point to move to the left. In order to confirm its practical application ability in the medical field, the rGO FET-based biosensor was used to detect serum samples of healthy people and prostate cancer (PCa) patients, and the test results showed significant differences, as shown in Figure 6c. Unlike other techniques, it is an effective tool that can use exosomes as markers for the early detection of fluid biopsy diseases.

### 4.2. Graphene Microfluidic Optical Sensors

With the continuous improvement of microfluidic technology and the continuous penetration and integration with other disciplines, several research hotspots have emerged in recent years, of which microfluidic optical devices are typical representatives. A new cutting-edge cross-discipline formed by the combination of optoelectronics and microfluidics technology combines microfabrication technology with physics, chemistry, biology, etc., and realizes the special functions of optical or optoelectronic devices and systems through precise control of microfluidics. With the birth of the new discipline of microfluidic optics and the development of new technologies, microfluidic optics will play a more important role in the future of optical technology. In contrast to traditional optical systems with large volume, high cost, and poor adjustability, the fusion of microfluidic technology and optical devices provides the possibility of miniaturization, arraying, low-cost and high-precision control of microfluidic optical devices.

Integrated optics can accommodate the compact arrangement of microfluidic channels and optical devices, which has broad application prospects for the integration of fluid optical sensors with high sensitivity and high throughput. Various optical methods have been applied to microfluidic sensings, such as light absorption, Raman scattering or surface plasmon resonance, and other measurement methods. Besides, graphene possesses excellent optical properties, especially under the total internal reflection structure, the enhanced interaction between graphene and light and its polarization-dependent properties, making graphene combined with microfluidics technology demonstrate potential applications in optical sensing. Wu et al. [98] developed an intelligent optical microfluidic sensing system with reflection coupling structure based on rGO glass for ultra-sensitive real-time detection of microfluidic liquid (water) pressure. The sensor is based on the principle of evanescent wave coupling, that is, the liquid around the sensing layer graphene interacts with its evanescent field. The schematic diagram of the optical experimental platform based on the rGO microfluidic sensor is shown in Figure 7a, with the microfluidic chip in the illustration being the most important part of the entire sensing system. The materials composing it from top to bottom are microfluidic channel, rGO, glass sheets, refractive index matching fluids, and prism. The key to the entire assembly process of the rGO microfluidic chip is that the PDMS microfluidic channel is aligned and bonded with the patterned rGO glass, and further adhered to the prism coated with a refractive index (RI) matching liquid, as shown in Figure 7b. The microfluidic channel is made of PDMS prepolymer, and its chamber size is 6 × 4 × 0.05 mm^3^, and the diameter of the two ports connecting the channel is about 10 μm. Based on the theoretical analysis of the reflection of the coupling structure, the strong interaction between a part of the incident energy and rGO is very sensitive to the RI change of the low refractive index medium. Therefore, when the pressure of the aqueous solution in the microfluid changes, its RI will change accordingly, which is measured by the photodetector. In their work, a fixed frequency (1 MHz) weak ultrasonic wave was used to provide a pressure of 1kpa to the water, and the RI change of 1.35 × 10^−7^ was obtained by using the formula ($dn/dP=1.35\times 10^{-10}$) between RI change (*dn*) and ultrasonic pressure (*dP*) in the water. The RI of water changes periodically with the ultrasonic frequency. Figure 7c shows that the ultra-small water RI change of about 470 mV under the sound pressure of single pulse ultrasound, corresponding to a response time of 560 ns and a frequency of 1 MHz, which is similar to the original ultrasound frequency.

The limit of detection (*D*) and sensitivity (*S*) for graphene-based microfluidic optical sensors are determined by the methods described above. The relationship between them and RI changes is as follows:(1)$$D=N_{noise}/S$$
(2)S=dU/dn
where dU is the value of the voltage signal change corresponding to a RI change in the medium 2. The above Formulas (1) and (2) give a detection limit of 1.4 × 10^−8^ and a sensitivity of 3.5 × 10^9^ mV/RIU for this microfluidic optical sensor. To facilitate the reliability of the experiment and optimise the signal-noise ratio (SNR), the high RI changes in the aqueous solution caused by high-pressure ultrasound were also monitored using the microfluidic optical sensor (see Figure 7d), resulting in an ultra-fast response time (about 600 ns) and a low SNR (about 23). By comparing the pressure of different ultrasonic waves on water, it is found that the response time limit is about 100 ns. In this regard, the application of ultrasonic waves to the optical path not only makes the detection process more accurate, but also makes it easier to directly perform real-time measurement without complicated operations. Besides, to further accurately detect the voltage fluctuation caused by the accompanying RI change, the water pressure in the detection window is increased by changing the water level in the external hose. The inset of Figure 7a shows a schematic diagram of a specific experimental device with a ball valve installed at the outlet of the microfluidic channel to make it in a closed state. When the height of the water in the external hose at the entrance is changed, the microfluidic optical sensor based on rGO glass can measure the change of the voltage signal, as shown in Figure 7e. This means that the tiny but stable voltage caused by the liquid level in the hose is accurately detected in the detection region. As the water level increases, the water pressure also presents an approximately linear relationship. In short, this work proves for the first time the importance of microfluidic devices for the high-precision detection of small changes in fluid pressure, and opens up new platforms for monitoring small changes.

In addition to the study of fluid pressure sensing in microfluidic environments, real-time monitoring and sensing of changes in fluid flow velocity is also challenging. Wu et al. [99] have designed a graphene-based microfluidic flow sensor (GMFS) that monitors weak and transient signals of flow changes in the detection window. Compared with the fluid pressure sensing experiment, graphene produced by low-pressure chemical vapor deposition (LPCVD) was used instead of rGO as the sensing layer, and the entire microfluidic channel was unobstructed in this experiment. As graphene exhibits a difference in absorption between TE and TM polarisation modes under total reflection conditions, the interface of the high RI medium (prism)/graphene/low RI medium (liquid) coupled sandwich structure enhances the interaction of light graphene, making it sensitive to changes in low RI liquids. In this work, the continuous liquid flow rate is calculated by the relationship formula (v=dQ/dS) between the volume flow rate per unit time (dQ) and the cross-sectional area of the microfluidic cavity (dS).

The relation between the detected voltage variation (dU) and velocity, the pressure (dP) in microfluidics and the actual fluid velocity signal (dv) are all nearly linear:(3)$$dU/dv=4.65\times 10^{5}\;mV\cdot sm^{-1}$$
(4)$$dP/dv=9.43\times 10^{5}\;Pa\cdot sm^{-1}$$
where the linear relationship between pressure and fluid velocity variation is obtained by COMSOL simulation. Since both pressure and flow rate has a linear relationship with voltage, the trend of voltage signal and pressure is considered to be similar to the trend of flow rate. In the experiment, the response of GMFS to the rapid change of the liquid flow rate provided by the syringe and the peristaltic pump was studied separately. Figure 7f shows the real-time response signal results caused by weak pressure detected by the graphene-based microfluidic optical sensor. It can be seen from the figure that the transient pressure given by the outside world will get a high periodic voltage signal, which corresponds to the signal change at each time point. Besides, the dynamic images of the theoretical simulation process intuitively illustrate the change of fluid velocity in the microfluidic channel. Based on the difference between the contraction and relaxation of the right ventricle of the heart, a peristaltic pump is used to simulate its cardiac cycle. The simulation experiment results cover the entire range of normal values, as shown in Figure 7g,h. The result in Figure 7g is that the voltage signal detected by GMFS corresponds to a peristaltic pump with a pressure of 3 kPa and a frequency of 1 Hz. Figure 7h shows that under low pump frequency, the stability of the simulated flow rate increases, and the noise of the peristaltic pump decreases, which corresponds to a peristaltic pump that provides a pressure of 11 kPa and a frequency of 0.1 Hz. The detection signal of the simulation process is far beyond the normal level, which indicates that GMFS can detect the weak abnormal signal in the blood flow velocity caused by the body organs, like the heart.These signals can reflect vascular occlusion, heart function and other basic vital signs, and also provide a possible new monitoring method for life medicine.

### 4.3. Graphene Microfluidic Biosensor

In the past decade, various nanomaterials combined with microfluidic chips have been developed to design sensors for detecting biomolecules. The appearance of graphene and its oxidized derivatives (such as GO, rGO, etc.) has become a vital nanomaterial in the field of biosensors due to its unique optoelectronic properties, especially in the manufacture of low-cost optoelectronic devices. The channel size and flow characteristics of microfluidic devices have also become favorable conditions for the study of biosensors and biological systems. In the field of sensing and biosensing, the research goal of combining microfluidic technology is to develop integrated miniaturized equipment that can achieve high sensitivity and selectivity, fast response, small throughput, and automated testing.

The combination of microfluidic technology and biosensing provides the advantages of high-throughput analysis and portability, thereby realizing intelligent real-time detection. Due to the different sizes of biomolecules, graphene nanomaterials can be used to functionalize and detect different biological analytes [100], such as nucleic acid molecules, protein molecules, bacterial cells, etc. Liu et al. [101] developed a graphene transistor sensor modified by DNA molecules in a microfluidic channel to sensitively detect BPA. In this work, a microfluidic chip was fabricated by patterning the channel in PDMS using a soft lithography machine, corresponding to the channel length and width of 10 mm and 0.8 mm, respectively. Figure 8a shows the structure of the entire microfluidic sensor, including the patterned graphene film under the drain-source electrode, and the bonding of the PDMS chip and the glass substrate through oxygen plasma treatment. In the whole process of detecting BPA, the graphene channel is first modified with Au NPs, and then the single-stranded probe DNA molecule is immobilized onto the Au NP on the surface of the graphene. After that, the BPA solution dissolved in the PBS solution with a concentration ranging from 10 ng/mL to 100 μg/mL was sequentially injected into the microfluidic channel to monitor the real-time response of the device with functionalized DNA to various concentrations of BPA in the PBS solution. It can be seen from Figure 8b that as the BPA concentration gradually increases from 1 to 100 μg/mL, the value of I_d_ shows an upward trend in the range of 55 to 62 μA. Besides, even at concentrations as low as 1 μg/mL, a significant current response to BPA injection can be observed. Due to BPA molecules in a high-concentration solution can form covalent compounds with DNA molecules, which destroy the DNA structure and chemical bonds, resulting in ssDNA molecules to fall off the graphene surface, as shown in Figure 8c. It also exhibits a unique signal response to the detection of BPA using double-stranded complementary DNA (dsDNA) molecules. The sensor provides a cost-effective way to detect BPA concentration in aqueous solutions with high sensitivity and is also expected to be used for convenient BPA detection in many practical applications. Singh and colleagues have fabricated a surface plasmon resonance (SPR) microfluidic biosensor that uses L-cysteine–reduced graphene oxide (L-Cys–rGO) hydrogel to quantify cardiac myoglobin (cMb), as shown in Figure 8d [102]. Figure 8e shows the fabrication process of this microfluidic biosensor. Cys-rGO hydrogel is covalently bonded to antibodies by introducing amide bonds. Microfluidic channels of micrometer dimensions are prepared and combined with glass containing Au, Cys-rGO hydrogels and Ag/AgCl electrodes by oxygen plasma catalysis. In order to use this microfluidic sensor to monitor cMb immunoassay based on the antibody-antigen interaction, the carboxyl-COOH functional group existing on the surface of Cys-rGO must be conjugated with the amine (-NH_2_) group of an antibody (cMAb). After injection into the microfluidic channel, cMb molecules are directly trapped by cMAb on the channel surface (see Figure 8f), leading to the formation of antigen-antibody immune complexes, which are detected by dual-mode transduction such as electrochemistry and surface plasmon technology. The large surface area and special structure of Cys-rGO hydrogel itself and the functional groups that can carry more antibodies on the surface provide favorable conditions for improving detection sensitivity and high selectivity. Therefore, the microfluidic biosensor not only shows a large dynamic detection concentration range of cMb, but also has high sensitivity and specificity. Besides, exploring its practical application in monitoring other biomolecules is the current trend of studying microfluidic biosensors.

Singh et al. [103] prepared a microfluidic immunosensor with high sensitivity to detect Salmonella typhimurium bacterial cells. The biosensor is made by selectively depositing a colloidal solution of GO-carboxy-lated multiwalled carbon nanotubes (GO-cMWCNTs) composite material on a patterned indium tin oxide electrode, and sealing it with PDMS microchannels, as shown in Figure 9a. Among them, when the mass ratio of GO:cMWCNT is 1:10, the largest electrochemical response is produced, which is found to be the most suitable mass ratio. Besides, the height and width of the PDMS microchannel manufactured by soft lithography are 200 μm, and the length is 2.0 cm. A single inlet and an outlet are connected to the microchannel chamber by punching holes at the desired positions at both ends of the PDMS plate (see the real image in Figure 9b). The carboxylation (COOH) on the GO-cMWCNTs composite was activated using 1-ethyl-3-(3-dimethylaminopropyl) carbodiimide (EDC) and N-hydroxysuccinimide (NHS) covalent chemistry, and then the antibodies (StAb) was immobilized to form discrete regions for capturing Salmonella typhimurium bacterial cells. With the injection of Salmonella typhimurium, the sensor shows the current change, as shown in Figure 9c. Due to the concentration of Salmonella typhimurium increases, an electrically insulating antigen-antibody complex is formed, which in turn leads to a decrease in the peak current at the sensor anode. It can also be found from the figure that the size of the electrochemical current varies linearly from 10^1^ CFU/mL to 10^7^ CFU/mL, and reaches saturation at 10^7^ CFU/mL. Compared with GO-based immunochips, the use of GO nanosheets to wrap cMWCNT exhibits a synergistic effect, which improves the sensing properties of Salmonella typhimurium cells and has better biological detection properties, such as sensitivity and detection limit. Figure 9d shows the electrochemical response comparison curve of only GO and GO-coated cMWCNTs-based bioelectrodes in the detection of Salmonella typhimurium. It can be seen that the detection limit and range of the sensor based on cMWCNTs wrapped with GO is significantly better than that of the sensor only with GO. The advantages of the microfluidic immunosensor with fewer reagents, higher sensitivity, reproducibility, and ease of functionalization and manufacturing are truly embodied in clinical studies on real samples of Salmonella typhimurium cells. In 2018, Jijie et al. [104] reported a sensitive and selective immunosensor for detecting pathogenic bacteria in drinking water and serum samples. Due to the formation of immune complexes, the detection of Escherichia coli is based on the limitation of electron transfer from redox mediators to rGO/polyethylenimine modified electric transducers. In a similar vein, Wibowo et al. [105] developed a graphene sensor as a means of detecting Escherichia coli in order to study the interaction between graphene and different concentrations of Escherichia coli. The sensor detects the signal response of Escherichia coli by monitoring the electrical properties and Raman spectroscopy of the graphene film. The increase of bacteria not only causes the resistivity of graphene to decrease, but also changes the intensity ratio between the G peak and the D peak. Therefore, the development of a device that can quickly detect harmful bacteria and is easy to implement is basically mature.

Here, more work is still needed to demonstrate the selectivity or specificity of the graphene microfluidic biosensor to the analyte. Ideally, it is more hope that the sensor can have a large response to the target detection object, but almost no response to any interference. In this regard, it is necessary to improve the functional layer, or more methods to capture the target.

## 5. Conclusions and Future Work

In this review article, we discussed the latest work on graphene and microfluidic technology, focusing on microfluidic sensors based on graphene nanomaterials for the detection of chemical substances and biomolecules. The combination of graphene and microfluidics can take advantage of the unique physical and chemical properties and analytical characteristics of graphene, such as real-time rapid detection, high sensitivity, low consumption, and easy operation. In recent years, more and more devices based on graphene integrate microfluidic have been developed and widely used in various fields. This article first introduces the preparation methods of graphene and its derivatives and the functionalization of graphene. Next, briefly introduce the development of microfluidic technology, and analyze the preparation of microfluidics and material research. It mainly discusses the manufacturing process of different types of microfluidic platforms and microfluidic devices. The main applications of graphene microfluidic devices are divided into three parts: electrochemistry, optics, and biology. It ranges from the detection of chemical molecules to the electrical signals of nucleic acid and protein molecules and even the detection of bacterial cells.

Although graphene microfluidics devices have made remarkable achievements in many fields such as biomedicine and health detection, research on graphene-based microfluidic sensors is still developing and improving, and there is much work to be done. In order to better use the graphene sensor, it can be converted into a sensing system that can be used in many daily life environments, such as drinking water quality monitoring, blood glucose level detection in the body, or as a urine analyzer. The main challenges for researchers in the microfluidics biosensors, in especial, using the graphene and turn them into production [86]. However, there lacked a method for obtaining the controllable, and easy preparation of graphene material with specific structure and properties [106]. Especially in the application of microfluidic biosensors, the main challenge for researchers is how to use graphene and put it into production. However, there is a lack of methods to obtain graphene materials with good structure and properties that are controllable, reproducible, scalable, and easy to prepare. Therefore, to obtain high-quality graphene-based nanomaterials, it is essential to find an effective method to grow graphene [107,108]. Many applications currently reported are limited to the detection of biomolecules, so new research needs to be extended to medical treatments. This also requires researchers to create new biosensors for different complex environments [109,110]. Besides, there is a need to deal with the toxicity and sustainability of graphene microfluidic devices. Because many graphene-based microfluidic devices are used for careful analysis, integrity and reproducibility must undergo extensive quality checks, which may increase costs. Therefore, there still faces great challenges to realize the production of cheap and environmentally stable graphene microfluidic equipment. But graphene microfluidic chip will eventually be the strongest candidate for one of these real-world tools. Through the organic combination of graphene and microfluidic, advanced manufacturing methods will be further developed and in the coming period, we will successfully implement various applications. This work could provide a reference for those who want to make research in the direction.

## Figures and Tables

**Figure 1 micromachines-11-01059-f001:**
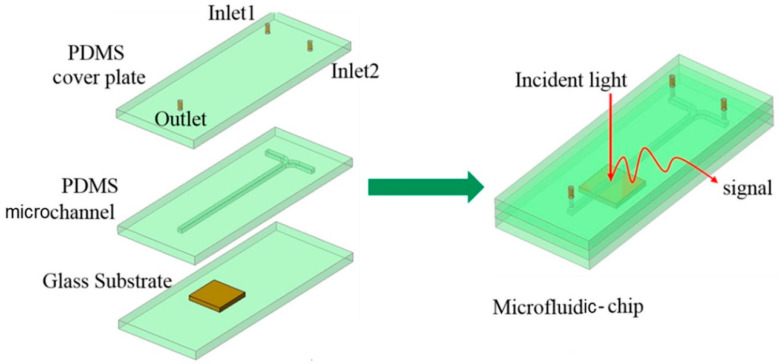
Schematic diagram of the microfluidic-chip. Reproduced from [17] with the permission from Elsevier.

**Figure 2 micromachines-11-01059-f002:**
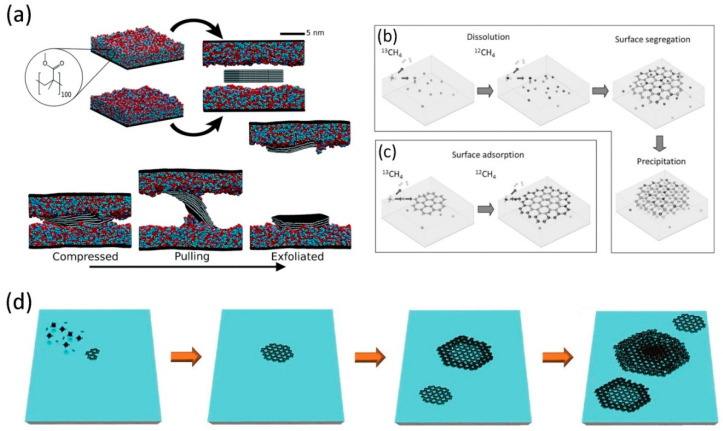
(**a**) Schematic diagram of direct exfoliation to obtain graphene by a top-down method. Reproduced from [49] with the permission from Royal Society of Chemistry. Mechanistic diagrams for the growth of graphene films by surface segregation (**b**) and surface adsorption (**c**). Reproduced from [58] with the permission from Wiley Online Library. (**d**) Schematic representation of the growth process of graphene on silicon substrate using atmospheric pressure chemical vapor deposition (APCVD). Reproduced from [59] with the permission from Open Access Science Online.

**Figure 3 micromachines-11-01059-f003:**
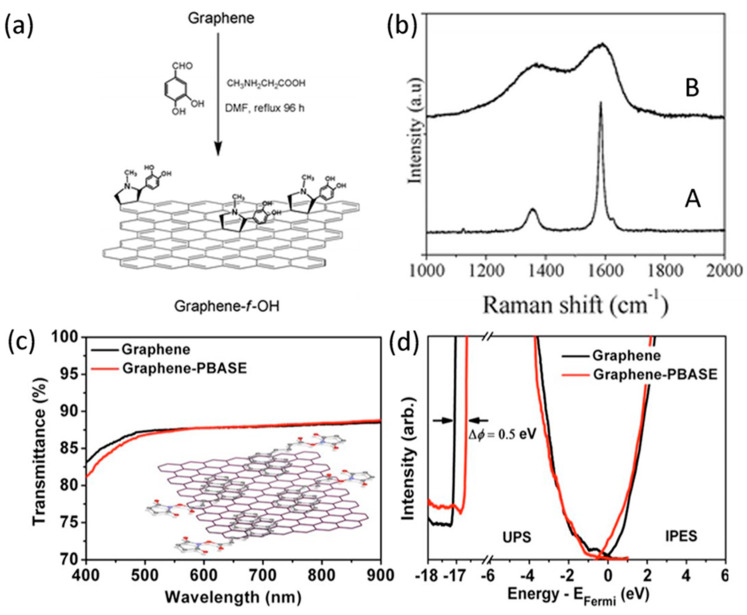
(**a**) Schematic diagram of modified functional groups on the surface of graphene. (**b**) Raman spectra of (A) starting graphene and (B) graphene-*f*-OH. Transmission spectrum. Reproduced from [65] with the permission from Royal Society of Chemistry. (**c**) and UPS and IPES combined spectrum. (**d**) of graphene film before and after modification. Inset in (**c**): schematic diagram of PBASE modified graphene. Reproduced from [66] with the permission from American Chemical Society.

**Figure 4 micromachines-11-01059-f004:**
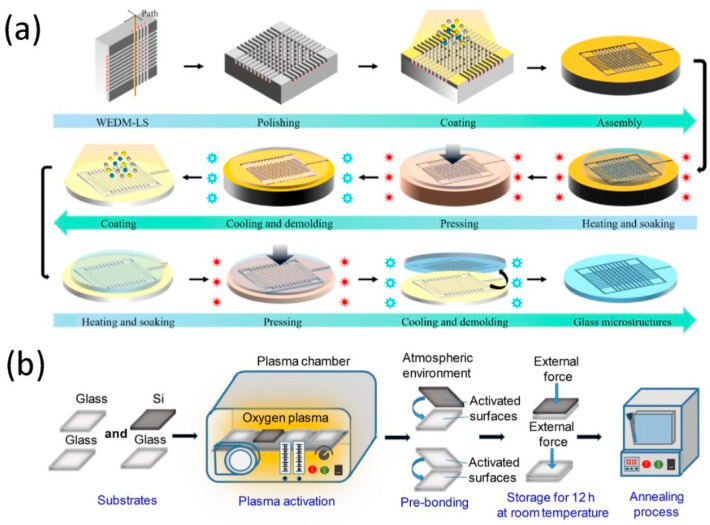
(**a**) Schematic diagram of the insert manufacturing process for tungsten carbide mold and high transition temperature glass mold; Reproduced from [74] with the permission from Elsevier Ltd. (**b**) Schematic diagram of oxygen-based plasma activation and low-temperature annealing process. Reproduced from [78] with the permission from Chinese Society of Metals.

**Figure 5 micromachines-11-01059-f005:**
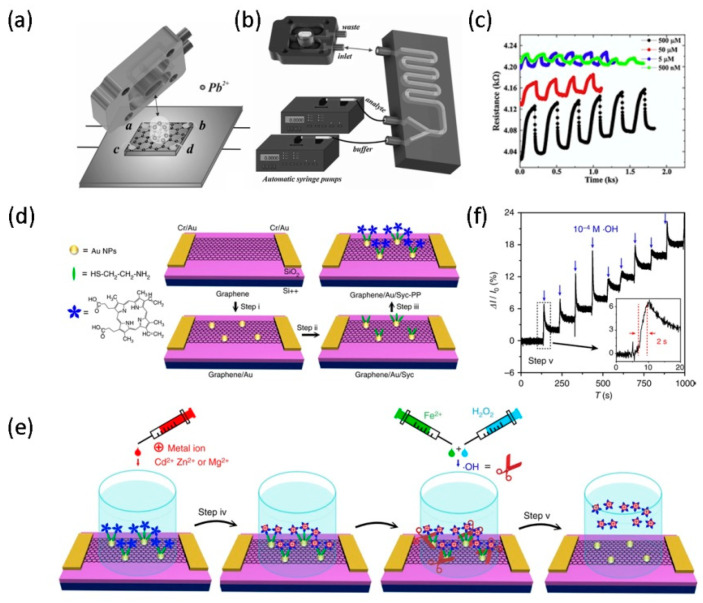
Diagram of the internal structure of the microfluidic chip (**a**) and the entire experimental equipment (**b**). (**c**) Sensor signals corresponding to different Pb^2+^ concentrations as a function of time. Reproduced from [41] with the permission from Elsevier. (**d**) Schematic diagram of the manufacturing process of a field-effect transistor (FET) sensor. (**e**) Schematic diagram of FET sensor •OH detection. (**f**) the real-time response signal of the microfluidic FET sensor after adding •OH. Reproduced from [96] with the permission from Nature Publishing Group.

**Figure 6 micromachines-11-01059-f006:**
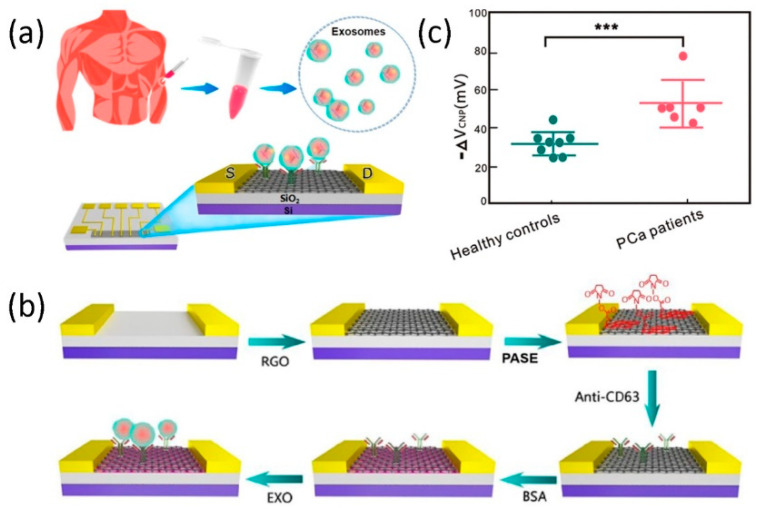
(**a**) Schematic illustration of the rGO FET-based sensor for the detection of exosomes. (**b**) The diagram of the process of detecting exosomes through rGO FET-based sensors. (**c**) The expression levels of different exosomes in serum correspond to healthy individuals and cancer patients. Reproduced from [97] with the permission from American Chemical Society.

**Figure 7 micromachines-11-01059-f007:**
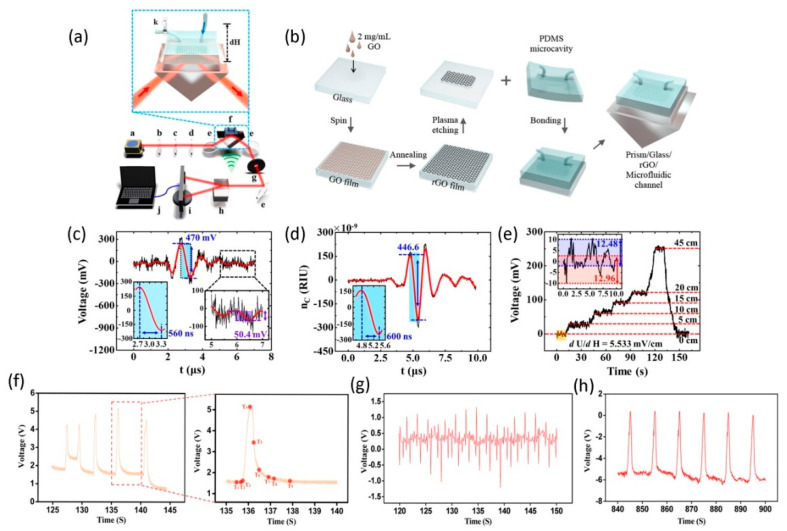
(**a**) Diagram of the experimental setup for the rGO-based microfluidic optical sensor. (**b**) The flow chart of the preparation for microfluidic optical sensing chips. (**c**) The weak RI variation of the solution corresponds to the pressure provided by ultrasound. (**d**) The optimized RI changes the detection of water. (**e**) The relationship between the liquid level in the catheter and the voltage signal response. Reproduced from [98] with the permission from Frontiers Media S.A. (**f**) The voltage-time signal collected from the video, the illustration shows the voltage-time signal between 135 and 140 s. The voltage and time signals corresponding to the liquid being pumped at higher (**g**) and lower (**h**) speeds, respectively. Reproduced from [99] with the permission from The Optical Society.

**Figure 8 micromachines-11-01059-f008:**
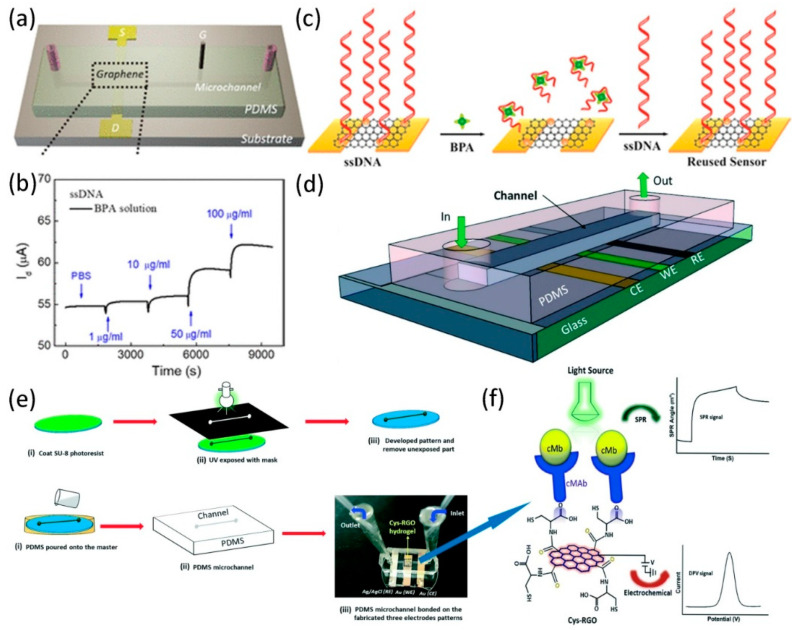
(**a**) Schematic diagram of a functionalized graphene transistor biosensor. (**b**) The time-dependent I_d_ for different concentrations of bisphenol A (BPA) solutions. (**c**) Schematic representation of the ssDNA molecule detaching from the graphene surface and binding again. Reproduced from [101] with the permission from American Chemical Society. (**d**) Schematic diagram of a microfluidic biosensor chip for cMb detection. (**e**) Flowchart for the preparation of Cys-rGO hydrogel microfluidic biosensor chips. (**f**) Cys-rGO microfluidic biosensor sensing mechanism and its detected signal response. Reproduced from [102] with the permission from American Chemical Society.

**Figure 9 micromachines-11-01059-f009:**
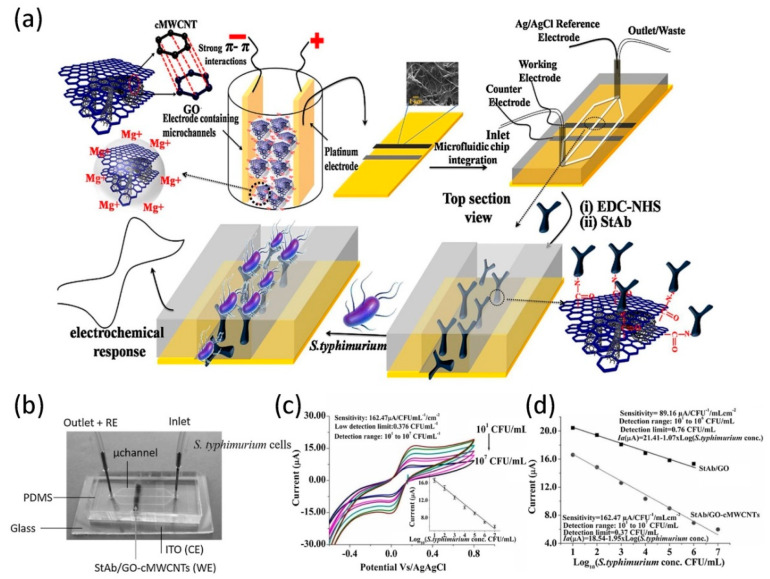
(**a**) Schematic for the fabrication of GO wrapped multiwalled carbon nanotubes integrated microfluidic chip. (**b**) Real image of the StAb/GO-cMWCNTs based microfluidic immunochip. (**c**) Electrochemical response of StAb/GO-cMWCNTs microfluidic electrodes to pathogenic cells. (**d**) Comparative electrochemical response curves of GO and GO/cMWCNTs based electrodes for Salmonella typhimurium detection. Reproduced from [103] with the permission from ELSEVIER.

**Table 1 micromachines-11-01059-t001:** The difference between graphene covalent functionalization and noncovalent functionalization.

Graphene Functionalisation	Theory	Bonding Types	Applications	Reference
Graphene covalent functionalization	Although the main part of graphene is composed of stable six-membered rings, the edges and the defects of it have high reactivity, it is easy to obtain GO by chemical oxidation method. There are abundant carboxyl, hydroxyl and active groups such as epoxy bonds which can be functionalized covalently by a variety of chemical reactions	Free radical addition atomic radical addition cycloaddition nucleophilic addition electrophilic substitution reactions	Polymer composite material, photoelectric functional materials and devices, biomedicine	[63]
Graphene noncovalent functionalization	The noncovalent functionalization of graphene utilize vdW and ionic interactions between graphene and a functionalized molecule	π-interaction(H-π interaction, π-π interactions, cation-π interaction, π_cation_-π interaction) electrostatic, hydrophobic, vdW interactions		[64]

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
