# Peer review of "Recent Advances in the Fabrication and Application of Graphene Microfluidic Sensors"

_micromachines, 2020, doi:10.3390/mi11121059_

Round 1

Reviewer 1 Report

Very nice review paper.

Author Response

Special thanks to you for your good recognition.

Reviewer 2 Report

This review refers to the progress and the recent developments of graphene-based microfluidic sensors. It is concise and well written.

Moderate English language and style changes are required. For example please check egain the sentences in lines 62-66, 110-111, 117-118, 180-184 etc.

Author Response

  1. Moderate English language and style changes are required. For example please check egain the sentences in lines 62-66, 110-111, 117-118, 180-184 etc.

Answer:

Thanks for reviewer’s kind suggestion. We have made correction according to the Reviewer’s comments. The sentences in lines 62-66, 110-111, 117-118, and 180-184 have been moderately changed.

 Special thanks to you for your good comments. 

This manuscript is a resubmission of an earlier submission. The following is a list of the peer review reports and author responses from that submission.

Round 1

Reviewer 1 Report

Wu et al presented a reviewing manuscript entitled “Graphene Microfluidics Biosensors” which, unfortunately, cannot be considered for a publication in Micromachines and the reasons are given in the following.

i) Although, to the best of my knowledge, there is not a similar review focusing specifically on three combined key-words from the title, the recent review by J. Sengupta and C. M. Hussain entitled “Graphene and its derivatives for Analytical Lab on Chip Platforms” published last year on TrAC (1016/j.trac.2019.03.015) is significantly more complete and better organized than the submitted manuscript. Just to exemplify, Sengupta and Hussain present an introduction centered in the topic (lab of a chip), sections and subsections on the fabrication of microfluidic devices for this purpose, (with different available technologies discussed), a section on the graphene production and functionalization, and finally, the applications. This is somehow what I expected from the submitted review paper by Wu et al, which unfortunately is not present.

ii) The manuscript’s spelling is far from appropriate. Not only the text is full of grammar errors, but the sentences are also poorly written, which makes them very often not clear. The way the sentences are placed and the ideas are organized are frequently confusing, which makes the reading not exciting. There are also some sentences that are meaningless. Finally, there are some typos spread out in the text, which some of them are unacceptable (e.g. lab-on-a-ship).

iii) From the title, a reader may expect to find in such a review the accomplishments and current challenges on the incorporation of graphene related-materials into microfluidic platforms for novel and/or improved biosensing applications. What I found tough, is a compilation of the literature cited only for the sake of exemplification, without going deep in the details related to the main theme of the proposed review. For example, nothing is said about the viability of either incorporating graphene-based materials into microfluidics or building microfluidics around graphene devices. Additionally, the authors do not discuss the behavior of graphene devices under flow or confinement. Finally, a review paper focused on microfluidics cannot afford to have the first illustration of a microfluidic device only in figure 4.

iv) As I somehow pointed out in (i), the sections here are not appropriate. The Introduction does not introduce the topic “Graphene Microfluidic Biosensing” but it is more a section dedicated to the (well-known) properties of graphene. Obviously, one or two paragraphs could be used to highlight the appealing characteristics of graphene, however, it would be desirable to highlight such properties in spite of the microfluidic biosensing applications. Additionally, for this section, the authors chose to illustrate graphene an AFM image of a flake and the graphene’s band diagram, i.e., no illustration related to “Graphene Microfluidic Biosensing” as should be expected for a section named “Introduction”. The other sections are a bit confusing as well. For example, for graphene electrochemical biosensors the authors present information on field-effect transistors (a consolidated technology that the authors call “very promising” on page 4). However, the working principle of FET sensors is not discussed. The authors simply present the structure of a FET (source, drain, gate, etc) and how to operate them (VGS and VDS application). There is no clear correlation to the information from this (sub)section with microfluidics applications using such a technology.

v) The selected figures are also questionable, as most of them are used just to illustrate literature examples, rather than the general concepts/technology behind graphene microfluidic biosensors. The figure quality is also not appropriate.

vi) The excess of references (156 papers) indicates that the authors could not select the most appropriate literature to be discussed. I wonder if the authors have the knowledge and, therefore, “scientific authority” to speak about graphene microfluidic biosensors.

For all the above-mentioned reasons I do not encourage the acceptance of such a review paper.

Reviewer 2 Report

I find this paper is really similar to the authors' previous Review article published at IJMS (MDPI). https://www.mdpi.com/1422-0067/20/10/2461/htm

Poor English.